# A Targeted UHPLC-MS/MS Method Validated for the Quantification of Ergot Alkaloids in Cereal-Based Baby Food from the Belgian Market

**DOI:** 10.3390/toxins13080531

**Published:** 2021-07-29

**Authors:** Bart Huybrechts, Svetlana V. Malysheva, Julien Masquelier

**Affiliations:** Sciensano, Rue Juliette Wytsman 14, 1050 Brussels, Belgium; Svetlana.Malysheva@sciensano.be (S.V.M.); Julien.Masquelier@sciensano.be (J.M.)

**Keywords:** ergot alkaloids, validation, UHPLC-MS/MS, baby food, QuEChERS

## Abstract

Following pending new legislation in the European Union setting a maximum of 20 ng g^−1^ for the total sum of ergot alkaloids in dry cereal-based baby food, a new UHPLC-MS/MS method was developed. It is suitable for the quantification of six ergot alkaloids: Ergocornine, ergocristine, ergometrine, ergosine, ergotamine, α-ergocryptine, and their corresponding epimers. The method is able to reliably detect individual ergot alkaloids at a level as low as 0.5 ng g^−1^. The method uses a modified QuEChERS extraction approach before UHPLC-MS/MS analysis. The method showed good sensitivity, accuracy, and precision. It has been applied to 49 samples from the Belgian market. In 26 samples, not a single ergot alkaloid was detected while in 23 out of 49 samples at least one ergot alkaloid was detected with 2 samples containing 12 ergot alkaloids. Ergometrine was the alkaloid most frequently detected i.e., 16 out of 49 samples. Only one sample, testing positive for all 12 ergot alkaloids, would be non-conforming to the newly proposed Maximum Residue Level (MRL).

## 1. Introduction

Ergot alkaloids are mycotoxins produced by fungi of the *Claviceps* genus and most notably by *Claviceps purpurea* [1]. Infections are most prevalent in cereals and wild grasses [2]. Rye is known to be especially susceptible, but wheat, barley, oats, and other cereal grains are also potential fungal hosts [3]. After the infection is established, the fungus replaces the developing grain or seed with an alkaloid-containing hard black tuber-like wintering structure, called a sclerotium. The sclerotium is often referred to as ergot or an ergot body. Both the alkaloid pattern and alkaloid contents may vary widely, due to differences in the maturity of the sclerotia, the fungal strain, the host plant, the geographical region, and the prevailing weather conditions [4,5]. As the sclerotia are harvested together with the cereals, this can lead to contamination of cereal-based food and feed products with ergot alkaloids. During the Middle Ages, the consumption of grains that were highly contaminated with ergot alkaloids was the cause of Holy Fire or St. Anthony’s Fire. Lower doses can induce symptoms including abdominal pain, vomiting, burning sensation of the skin, insomnia, and hallucinations.

German research has indicated an increase in the occurrence of *Claviceps purpurea* infections in the last few years possibly associated with the more extensive use of hybrid varieties of rye [3]. Although nowadays, effective cleaning procedures at mills allow to remove up to 82% of ergots from grain, ergot alkaloids are still being detected, [3,4,6,7]. 

Ergot alkaloids are a group of more than 40 indole alkaloids, commonly with an ergoline ring with a nitrogen atom at position 6 and substituted at C-8 and methylated at N-6 (see Figure 1). 

They have a double bond between either C-8 and C-9 or C-9 and C-10. The neutral ergot alkaloids are mainly derivatives of lysergic acid, with the six major alkaloids ergometrine, ergosine, ergocornine, ergocryptine, ergotamine, and ergocristine, also present as their isomeric -inines [3,8] (see Figure 2).

Ergot alkaloids with a double bond between C-9 and C-10 undergo epimerization with respect to the center of symmetry at C-8, resulting in rotating (C8-(S) configuration) isomers (see Figure 3).

The left rotating (C8-(R) configuration) epimers are referred to as ergopeptines and the right rotating (C8-(S) configuration) epimers are named ergopeptinines. These epimers differ in physicochemical properties allowing them to be separated relatively easily in liquid chromatography. C8-(R)-isomers (-ines) are biologically active, whereas the C8-(S)-isomers (-inines) are inactive [3]. The conversion of the -ine to the -inine isomers occurs rapidly in aqueous solutions and both forms are found together in naturally contaminated samples [1]. It is, therefore, crucial to consider both epimers when the contamination level of a cereal-based product has to be determined.

Different methods have been reported for the analysis of ergot alkaloids. Chromatographic and mass spectrometric methods to determine lysergic acid diethylamide (LSD) and related compounds in body fluids have been reviewed by Reuschel et al. [9]. However, validated LC-MS methods for the simultaneous quantitative determination of the major ergot alkaloids in food, and particularly baby food, are scarce [10,11,12,13,14]. 

Two decades ago, a maximum level of 0.05% *Secale cornutum* was specified in regulation 2000/824/EC [15] for durum wheat, wheat, and rye (bread grain or cereals). According to regulations 2003/1784/EC [16] and 2005/1068/EC [17], the intervention on rye has been suspended. There exists a maximum value of 0.05% for *S.* c*ornutum* in durum wheat and wheat according to regulation 2009/1272/EC [18]. In regulation 2015/1940/EU [19], a maximum level of 0.05% *S.* c*ornutum* is specified for certain unprocessed cereals except for corn and rice. For processed rye and rye flour, a maximum value of 0.05% *S. cornutum* is specified. Ergot alkaloids are considered to be mycotoxins, so maximum levels should be laid down in regulation 2006/1881/EC [20]. However, individual ergot alkaloids are not regulated for grain-based food. Nevertheless, to limit the health risk, the European Union intended to minimize and regulate the total ergot alkaloid content of cereals and marketable food for a long time. Monitoring ergot alkaloids in feed and food is strongly recommended by 2012/154/EU [21] and by regulation 2015/1940/EU [19]. In the EU, there is a discussion on setting possible maximum levels for ergot alkaloids on the sum of the following 12 ergot alkaloids: Ergometrine, ergosine, ergocornine, ergotamine, ergocristine, ergocryptine (α- and β-forms), and their respective -inine forms. The consensus would be that the –ine and –inine do not have to be individually determined, in other words, determining the sum of the –ine and –inine would suffice for enforcement reasons. For processed cereal-based baby food, the maximum residue level (MRL) would be set at 20 ng g^−1^ for the total sum of ergot alkaloids so a limit of quantification not higher than 1 ng g^−1^ per individual alkaloid would be preferable. The β-form of ergocryptine was not added to this method as its reference standard was not available at the time the method was initially developed. 

A QuEChERS-based method was first applied to ergot alkaloids by Malachova et al. [22]. Recoveries of ergocornine, ergocristine, ergocryptine, and ergosine ranged from 60% to 70%. The alkaloids were not detected in 116 cereal samples. A QuEChERS procedure was optimized for the extraction of ergovaline from tall fescue seed and straw for subsequent separation and determination by LC-FLD by vortexing the sample with ammonium carbonate/acetonitrile 1:1 (*v*/*v*) before adding magnesium sulfate and sodium chloride and vortexing again [23]. Following centrifugation, an aliquot of the separated acetonitrile phase was evaporated to dryness and the extract reconstituted in methanol. Mean recoveries ranged from about 90% to 98%.

The present study was designed to develop and validate a new UHPLC-MS/MS method that allows the simultaneous determination of the six ergot alkaloids defined by the European Food Safety Authority (EFSA) [24] as among the most common and physiologically the most active i.e., ergometrine, ergotamine, ergosine, ergocristine, ergocryptine, and ergocornine, together with all their corresponding -inine isomers. 

## 2. Results

### 2.1. UHPLC-MS/MS Method Development

#### 2.1.1. MS/MS Detection

The mass spectrometer was operated in the positive electrospray ionization (ESI+) mode. The critical MS parameters e.g., cone voltage, collision energy, precursor, and product m/z were optimized by infusing the integrated syringe pump with a 500 ng mL^−1^ standard solution containing a single alkaloid at 20 µL min^−1^ into a mobile phase running at 0.2 mL min^−1^ of 50/50 (*v*/*v*) mobile phase A and mobile phase B. The LC-MS was operated using MassLynx^®^ 4.1 software. MS parameters for the analysis were as follows: ESI source block and desolvation temperatures: 150 °C and 450 °C, respectively; capillary voltage: 0.5 kV; argon collision gas flow: 0.15 mL min^−1^; cone nitrogen and desolvation gas flows: 150 and 1000 L h^−1^, respectively. For all alkaloids, the proton-adduct [M+H]^+^ was chosen. Three MRM transitions were selected for each of the analytes. Detailed descriptions of the known and postulated fragmentation routes have been provided in the literature [25,26,27,28]. An overview of the parameters used in this method can be found in Table 1.

#### 2.1.2. MS/MS Detection

Ergot alkaloids are basic (alkaline) compounds containing a nitrogen atom that is not protonated in alkaline conditions. The non-protonated form is the preferred form for the chromatographic separation as this will result in higher retention times and thus such compounds will elute at higher percentages of the organic modifier. Additionally, the non-protonated form will be less susceptible to secondary interactions on the C18 column resulting in better peak shapes i.e., less tailing and higher S/N. In LC-MS, all of this will facilitate evaporation and ionization in the ESI source resulting in better sensitivity. The only downside to using an alkaline buffered mobile phase at ±pH 10 is that most silica-backboned C18 columns are not suitable for these conditions so a specific alkaline mobile phase-stable column was chosen.

An example of a chromatogram of a sample artificially spiked at 0.5 ng g^−1^ for each ergot alkaloid can be found in Figure 4.

### 2.2. Optimization of Sample Preparation

Ergot alkaloids are susceptible to epimerization under acidic and/or aqueous conditions [14,29]. In 2014, our group developed a method to quantify pyrrolizidine alkaloids at the pg g^−1^ level with no specific clean-up (e.g., SAX-SPE columns), which was based on an aqueous highly acidic extraction solvent [30]. For ergot alkaloids, this solvent is unsuitable due to the low pH causing significant and almost instantaneous epimerization. Therefore, two new approaches were tried this time with (1) a modified QuEChERS approach using an alkaline acetonitrile/water mixture and (2) aqueous methanol (MeOH/water/formic acid, 60/39/1, *v*/*v*/*v*). The first approach gave the best results by far: In this modified QuEChERS approach, 4g of the sample is extracted for at least 1 h with a 30 mL solution of 84% acetonitrile and 16% (*v*/*v*) water containing 200 mg L^−1^ ammonium carbonate. 

The alkaloids are non-protonated under alkaline conditions and will migrate more easily to the upper organic layer. As the upper acetonitrile layer has been separated from the water, this should also minimize epimerization. 

### 2.3. Method Validation

The method was validated following a 3 × 3 × 3 design: On 3 different days, a blank sample was spiked at three levels i.e., 0.5 ng g^−1^, 1 ng g^−1,^ and 2.5 ng g^−1^ alongside three blank samples (i.e., <LOD, Table 2). Each level was repeated three times resulting in 12 samples each day or 36 samples over 3 days in total. Additionally, each extract was spiked at 10 ng mL^−1^ to determine the matrix effect of the sample matrix. The final result for each sample was corrected for the matrix effect. No correction was made for recovery, in other words, the recovery was set at 100%. Documents 2002/657/EC [31] and 401/2006/EC [32] were used as guidelines for method validation. Raw data were processed using TargetLynx 4.1 software. 

For a positive identification of ergot alkaloids in the samples, three criteria needed to be met:

The concentration needs to be at least as high as the LOD.

The deviations of relative ion intensities for the MRM transitions should not be greater than the maximum permitted tolerances [31]. The relative ion intensities expressed as a percentage of the intensity of the most abundant ion must correspond to those of the ions in the solutions of standards and the recovery sample, with a maximum permitted deviation of 20% (relative ion intensity >50%), 25% (relative ion intensity: 21–50%), 30% (relative ion intensity: 11–20%), and 50% (relative ion intensity <10%).

The relative retention times with regard to the standards and QCs should not be higher than the maximum permitted deviation of 2.5%. 

#### 2.3.1. Analytical Selectivity and Carry-Over

Selectivity and carry-over were verified by two criteria: (1) The absence of detectable peaks of ergot alkaloids in a blank extraction solvent sample injected directly after the highest calibration point and (2) the ion ratio and retention time stability relative to the solvent-based standards and QCs in the spiked samples as defined by 2002/657/EC [31] i.e., a maximum 2.5% difference in the retention time within one run and ion ratio variability.

When running a blank sample after the highest calibration point, no signal was detected for any of the 12 epimers.

The ion ratios on a sample artificially spiked at 0.5 ng g^−1^ (Figure 4) were not distinguishable from the ion-ratios on a solvent based standard or QC

#### 2.3.2. Recovery

Each individual sample was corrected for matrix effect. The signal was altered by matrix-induced enhancement rather than matrix suppression and varied between 91% and 161% with an average of 139%. After correction for matrix effects, the true recovery varied between 80 and 111% with an average of 103%.

#### 2.3.3. Linearity

Linearity was verified by examining (1) the R^2^ for each component and (2) the absence of a visually detectable pattern in de calibration point residuals. R^2^ observed for the calibration range of 0.02 to 100 ng mL^−1^ was always at least 0.99. No specific pattern was observed in the residuals e.g., a U-shape pattern, which could indicate, for instance, component adsorption on the hardware with the lower calibration points and/or signal saturation of the detector at higher concentrations. A first-order linear design weighted at 1/X was used for all components.

#### 2.3.4. Limits of Detection and Quantification and Reporting Limit

An LOD based on the 3.3* S/N and an LOQ based on the 10*S/N approach applied to the spiked recovery samples gave an indication of the intrinsic sensitivity of the method. These indicated that the method is sufficiently sensitive to measure 0.5-times the proposed MRL of 1 ng g^−1^ per alkaloid. The intrinsic LOD varied from 0.03 to 0.17 ng g^−1^ and the LOQ varied between 0.1 and 0.5 ng g^−1^ (Table 2). For future routine enforcement analyses, it was decided to use the lowest validated level as the limit of quantification. To differentiate between the latter value and the LOQ = 10* S/N, the lowest validated level is referred to as the “reporting limit” or RL for short.

#### 2.3.5. Precision and Accuracy

The total measurement uncertainty MU (k = 2) varied from 26% to 52% with an average of 37%, which for these low levels can be considered as satisfactory. The average bias, determined as the slope of a recovery linear regression in Table 2, was 5% and the bias values varied between −20% and 20%. Given that the total measurement uncertainty (MU) is around 40% and thus much higher than the bias, in other words with an uncertainty of 40%, it is not possible to determine whether the observed biases of max 20% are actually significant, and it was decided not to apply bias correction in this method. 

A detailed overview of the validation results per validation level can be found in Table 3. The bias per level fell within the +/− 20% range. Both the repeatability and the within laboratory reproducibility fell within the Horwitz limits. The Horwitz limits defined by RSD_R_ = 2^(1−0.5 log C)^ C, where C is expressed as a dimensionless mass fraction [31]. Only for ergometrinine at the lowest level was the repeatability marginally higher than that allowed by Horwitz i.e., 34% versus a maximum of 33%.

#### 2.3.6. Epimerization

Although the future MRLs in the European Union would be based on the sum of the –ine and –inine form, implying that epimerization is not critical as long as the sum of the –ine and –inine remains unchanged, it was decided that in our method, ergot alkaloids needed to be individually quantified.

Epimerization of ergot alkaloids is a process that is facilitated under acidic conditions in either protic organic solvents (e.g., methanol) or water. 

The conditions chosen for both the preparation of standards and the extraction, i.e., alkaline conditions combined with acetonitrile, should minimize epimerization. No discrepancies that could point to epimerization were detected during the method validation e.g., an unusual low recovery for an –ine alkaloid and an unusually high recovery for the–inine (or vice versa). This method was used to analyze food and feed samples during a proficiency test and our satisfactory results confirmed this.

### 2.4. Method Application for Analysis of Belgian Baby Food

As described in the “Samples” section, the validated LC-MS/MS method was subsequently used to investigate contamination of cereal-based baby food products that are available on the Belgian food market. Forty-nine samples were analyzed using the above-described method. Most samples consisted of dry cereal-based cookies, some based on rice, and some additionally flavored e.g., with fruit, chocolate, vegetables, or cheese. The other samples mostly consisted of some kind of porridge, unflavored or flavored. An overview of the raw data per sample can be found in Table 4. If the method will be run for enforcement reasons, the reporting limit of 0.5 ng g^−1^ will be applied, but for scientific publications, values will be higher than the LOD, given that the criteria for both the retention time and ion ratios are met.

## 3. Discussion

Our initial goal was to develop a method that would be able to simultaneously detect three types of alkaloids i.e., the pyrrolizidine alkaloids, tropane alkaloids, and the ergot alkaloids. It became clear that due to their different physico-chemical properties it would never be possible to develop a method that could detect all three of them and still meet strict validation guidelines. So, the choice was made to develop separate methods. More details on the method for pyrrolizidine alkaloids can be found in Huybrechts et al. [30].

The proposed method for ergot alkaloids was validated with a 3 × 3 × 3 validation scheme. All critical parameters i.e., selectivity, sensitivity, measurement uncertainty, and recovery (bias) were met without using any complicated clean-up procedure (e.g., concentration via an SPE step) making this method suitable for the detection and quantification of ergot alkaloids in dry cereal-based baby foods at the newly proposed sum MRL of 20 ng g^−1^. Although the maximum levels for the individual ergot alkaloids will be set for the sum of the –ine and –inine forms and thus epimerization during sample treatment would not be an issue for legal conformity testing, we opted for an extraction procedure that limits epimerization as much as possible.

Ergometrine was the ergot alkaloid most frequently detected (see Figure 5) i.e., 14 out of 49 samples of cereal-based baby food from the Belgian market followed by α-ergocryptinine and ergocorninine. The single highest concentration detected was for ergotaminine at 13.4 ng g^−1^.

In Figure 6, a graphic representation is given of the number of ergot alkaloids detected in a single sample. 

Twenty-six samples were not contaminated, while two samples contained all 12 ergot alkaloids. Out of these two samples, one sample, a ladyfingers dry, egg-based, sweet boudoir biscuit, contained a total of 42 ng g^−1^ and would be labeled non-conform. All other 48 samples were unambiguously conforming to the proposed MRL of 20 ng g^−1^.

The concentrations detected in these dry baby food samples were remarkably lower than the concentrations detected in food and feed on the Belgian market in another study [14].

## 4. Conclusions

To conclude, the present study describes a detailed validated method for the quantification of ergot alkaloids (-ine and -inine isomers) in cereal-based baby food. This method has excellent validation parameters i.e., specificity, linearity, recovery, repeatability, reproducibility, and measurement uncertainty, thus enabling the accurate quantification of six ergot alkaloids, ergocornine, ergocristine, ergometrine, ergosine, ergotamine, α-ergocryptine, and their corresponding epimers, and their epimers in baby food. A modified QuEChERS-type extraction-based method with UHPLC-MS/MS as a detection technique was presented. This method is suitable for high-throughput screening of baby food at a level as low as 0.5 ng g^−1^ per alkaloid. Satisfactory results from participation in a proficiency test confirmed that this method is fit for this purpose. Forty-nine commercially available samples were screened for their ergot alkaloid profile and only one sample was found to be non-conforming at a level (total sum of 12 ergot alkaloids) of 42 ng g^−1^. On the whole, the concentrations detected in dry baby foods were significantly lower compared to the ones detected in a previous study in food and feed available on the Belgian market.

## 5. Materials and Methods

### 5.1. Standards

Certified reference standards of ergot alkaloids (ergocornine, ergocristine, ergometrine, ergosine, ergotamine, α-ergocryptine, and their corresponding epimers) were purchased as dried down films from Romer Labs (Tulln, Austria). Standard solutions of each compound were produced by adding 5 mL ULC-MS grade acetonitrile to the vials following the instructions of the manufacturer, resulting in the following concentrations: The -ine epimers were all at a concentration of 100 µg mL^−1^, while the inine-epimers were all at 25 µg mL^−1.^. Three working solutions containing all 12 ergot alkaloids were prepared in pure acetonitrile: (1) A first working solution at a concentration of 1 µg mL^−1^, and this solution was used to prepare (2) a second working solution of 0.1 µg mL^−1^, and this was used to prepare (3) a third working solution at 0.01 µg mL^−1^. All solutions were stored at −20 °C as recommended by the vendor to minimize epimerization. Ergot alkaloid standards are best stored below −20 °C in non-protic solvents or in the form of thin dry films [33], which has been shown to be suitable for over 12 months [13].

#### 5.1.1. Reagents and Consumables

LC-MS grade ammonium hydroxide was purchased from Sigma-Aldrich (Overijse, Belgium), and ULC-MS grade ammonium acetate, ULC-MS grade acetonitrile (ACN), and HPLC-S grade ACN were from Biosolve (Valkenswaard, The Netherlands). The water was produced by a Milli-Q system (Millipore, Overijse, Belgium). Two-mL syringes, 50 mL PP (polypropylene) tubes, AR-grade ammonium carbonate, AR-grade magnesium sulfate (MgSO_4_), and AR-grade sodium chloride (NaCl) were purchased from VWR (Haasrode, Belgium). Two-mL amber glass vials were from Waters (Waters Corp., Milford, MA. USA). Lastly, 0.2 μm polytetrafluoroethylene (PTFE) syringe filters were from Phenomenex (Utrecht, The Netherlands).

#### 5.1.2. Instrumentation and UHPLC-MS/MS Conditions

The UHPLC-MS/MS system consisted of an Acquity UPLC^®^ H-Class coupled to a XEVO^®^ TQ-S mass spectrometer (both from Waters Corp., Manchester, UK), and the mass spectrometer was operated in ESI(+) mode. Experiments were carried out in the multiple-reaction monitoring mode (MRM). 

One microliter of the sample was separated on a Phenomenex Kinetex EVO C18 column (1.7 µm, 100 mm × 2.1 mm) fitted with a 0.2 μm prefilter from Waters. The column temperature was maintained at 40 °C. Mobile phase A consisted of an aqueous solution of 0.05 % ammonium hydroxide (*v*/*v*) and 5 mmol L^−1^ ammonium acetate. Mobile phase B consisted of pure ULC-MS grade acetonitrile, and no buffer was added to this phase. The use of a UHPLC system instead of an HPLC strongly reduces the chromatographic run time [34]. A linear gradient of mobile phase B starting from 10% to 95% in 7 min (±12% B/min) running at a flow of 0.5 mL min^−1^ was used to separate the components. The column was subsequently washed with 95% mobile phase B for 1 min and allowed to equilibrate to the start conditions for 2 min resulting in a total run time of 10 min.

### 5.2. Preparation of Calibration Standards and Quality Controls

#### 5.2.1. Calibration Standards

The mass spectrometer was calibrated each run by using solvent-based standards prepared at the following concentrations: 0.02, 0.05, 0.1, 0.25, 0.5, 1, 5, 10, 25, 50 and 100 ng mL^−1^, corresponding to the range from 0.126 ng g^−1^ to 630 ng g^−1^ in the sample (dilution factor extraction solvent-solid sample = 6.3). The standards were prepared by diluting the 1 µg mL^−1^ solution to a 100 ng mL^−1^ solution followed by sequential dilution to prepare the other standards. Acetonitrile was chosen as the solvent as this (1) mimicked best the sample extract composition and (2) the epimerization of ergot alkaloids is less of an issue in aprotic solvents.

#### 5.2.2. Quality Controls

Three different types of quality controls were used:

System Suitability Checks: At the beginning of each analytical run a solvent-based control with a concentration of 4 ng mL^−1^ was injected at least 3 times to verify that the LC-MS is working correctly i.e., (1) retention time variation is less than 1% between injections, (2) signal intensity variation is below 10% for 2 consecutive injections, and (3) the peak width of the first eluting compound, ergometrine, should be less than 0.1 min applying the 10% elution volume rule of thumb. This is repeated every six samples and at the end of each run. For the remainder of the run, the following criteria are used: (1) Retention time is sufficiently stable i.e., ≤2.5% between every 6th injection, and (2) the difference in response signal is not higher than 10% between consecutive injections and not more than 20% over the entire run.

Recovery sample: Each analytical run, a blank sample fortified to a level of 1 ng g^−1^ for each ergot alkaloid before extraction is analyzed following the same protocol as the unknown samples. The recovery should fall for each ergot alkaloid after correction for the matrix effect between 70 and 120%. The recovery is defined as the proportion of the amount of analyte added versus the amount of analyte detected after correction for the matrix effect i.e., true recovery, and is calculated as follows: Recovery (%) = Measured concentration/Theoretical concentration × 100. This sample is, in other words, used to verify whether the sample preparation step has worked.

Standard addition sample: No isotopically labeled internal standards were available on the commercial market when the method was developed, so another approach had to be chosen to correct for matrix-induced signal intensity alteration during ionization in the ESI source. Matrix-matched calibration was rejected in favor of a single standard addition point for each sample. Mulder et al. [35] modified his approach to quantify ergot alkaloids in cereal samples by using a duplicate analysis in which one subsample was spiked with ergot alkaloids to enable an approximate calculation of the concentration to be made. This is effectively a single-point standard addition. When this showed samples to contain a significant quantity of ergot alkaloids, they were then determined accurately using multi-level standard additions. The latter approach has 2 main advantages over a calibration curve prepared in a matrix extract: (1) Matrix effects can vary widely between individual samples. A calibration curve prepared in one arbitrarily chosen matrix can never correct for this and (2) in some matrices, the matrix-induced signal suppression can be so strong that the sample is labeled incorrectly as negative or conforming (false negative). It has been reported in the past that signal intensity was suppressed by the matrix more strongly for the early eluting compounds, with a significant difference between the grain types [35]. So, a single standard addition approach was chosen in which 45 µL of each sample extract was fortified with 5 µL of the 100 ng mL^−1^ spike solution resulting in a concentration of 10 ng mL^−1^. To correct for the dilution effect, each sample was diluted with the extraction solvent in the same ratio i.e., 45 µL of the extract with 5 µL of solvent.

### 5.3. Samples

Forty-nine samples of baby food were purchased in supermarkets and local, mostly organic, food stores in Belgium. All samples consisted of dry cereal-based baby food (e.g., cookies/biscuits, oatmeal baby cereal, etc.). No vegetable, fruit, meat-based, or infant formula samples were selected as the new proposed legislation does not apply to them. Samples were ground down using a Retsch knife mill grindomix GM 200.

### 5.4. Sample Preparation

Four grams (±0.02 g) of the sample was weighed in a disposable 50 mL PP centrifuge tube with a screw cap. Thirty milliliters of an aqueous alkaline extraction solvent consisting of 84 % (*v*/*v*) HPLC-S-grade acetonitrile, 16 % purified water, and 200 mg L^−1^ ammonium carbonate was added and the tube was shaken on a Reax 2 overhead shaker (VWR, Haasrode, Belgium) for 60 min. A mixture of 6 g MgSO_4_/1.5 g NaCl was added, vigorously shaken for 30 sec, and centrifuged shortly to complete the phase separation and remove most of the undissolved salts. Using a deliberate salt-induced phase separation, resulting in an upper acetonitrile phase in which the analytes are concentrated, has several advantages: (1) It makes the issue of an unwanted phase separation resulting in false high recoveries irrelevant and (2) the ergot alkaloids are concentrated in an aprotic solvent minimizing the risk of epimerization. The N-6 nitrogen gives the protonated ergot alkaloids pKa values of 5.0 to 7.4, they have a positive charge in acid solution, and are neutral in alkali [29]. Two milliliters of the organic layer was filtered through a 0.2 µM PTFE filter. Then, 45 µL of the extract was diluted with either 5 µL of acetonitrile or with 5 µL of the 0.1 µg mL^−1^ working solution resulting in a concentration of 10 ng mL^−1^ in the extract. The latter sample was used to verify and quantify the matrix effect for each sample. Recovery was verified by spiking 400 µL of the 0.01 µg mL^−1^ working solution on a blank (concentration ≤ LOD) sample matrix. The spiked sample was left for at least 30 min in the dark at room temperature to allow the solvent to evaporate. 

## Figures and Tables

**Figure 1 toxins-13-00531-f001:**
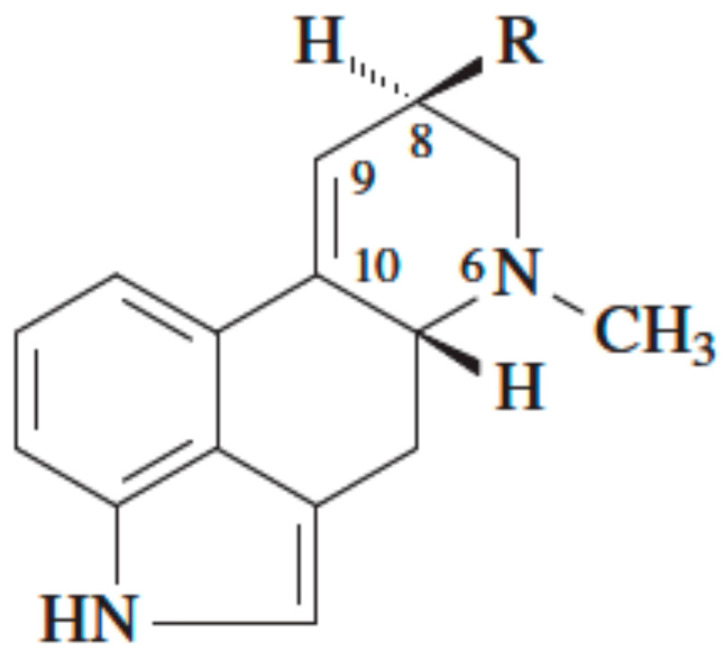
Structural backbone of ergot alkaloids.

**Figure 2 toxins-13-00531-f002:**
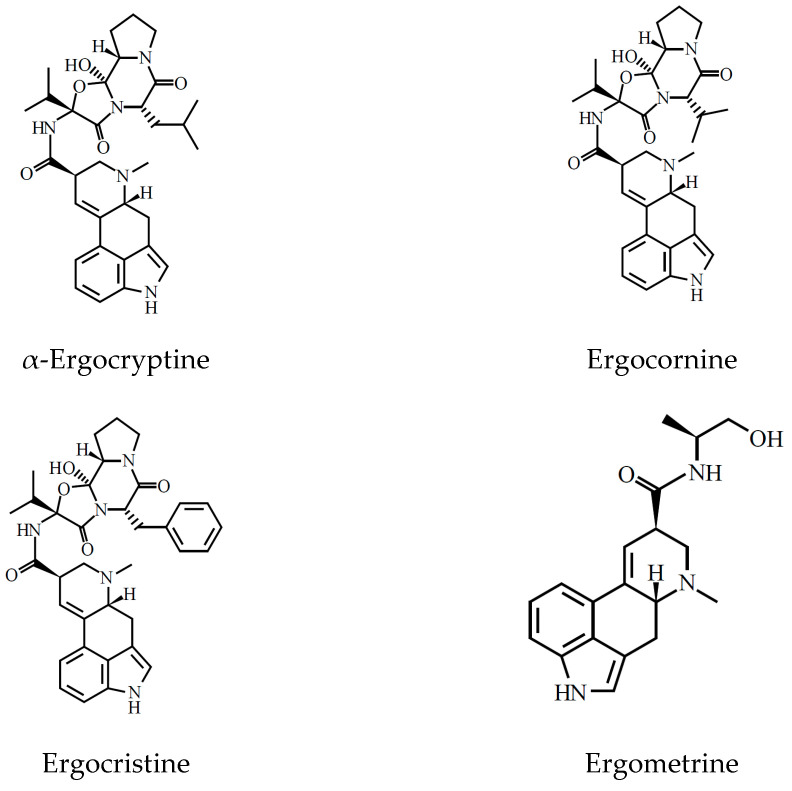
Structures of the individual ergot alkaloids.

**Figure 3 toxins-13-00531-f003:**
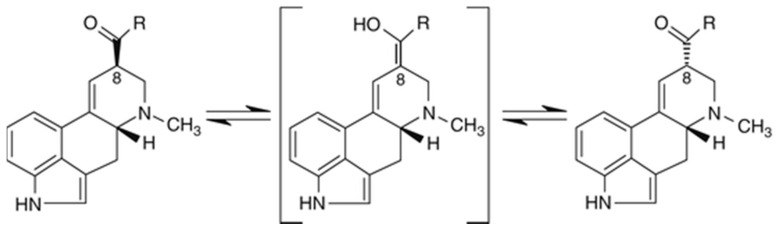
Example of epimerization at the C-8 position.

**Figure 4 toxins-13-00531-f004:**
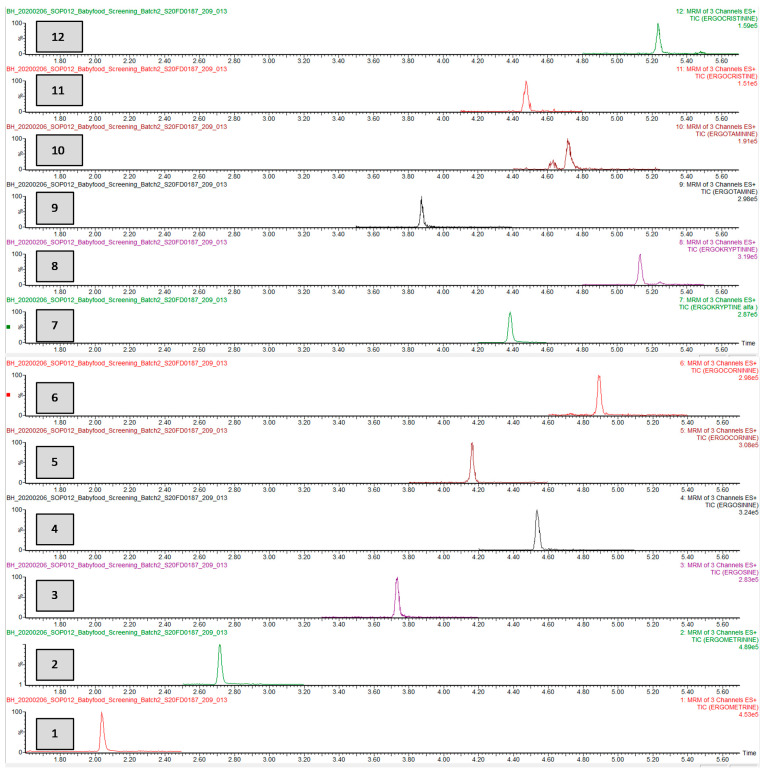
Chromatogram of a blank baby food sample spiked at 0.5 ng g^−1^. The ergot alkaloids shown are from bottom to top: (**1**) Ergometrine, (**2**) ergometrinine, (**3**) ergosine, (**4**) ergosinine, (**5**) ergocornine, (**6**) ergocorninine, (**7**) α-ergocryptine (**8**), α-ergocryptinine, (**9**) ergotamine, (**10**) ergotaminine, (**11**) ergocristine, and (**12**) ergocristinine. A small secondary peak originating from the matrix can be seen in the 10th channel of ergotaminine at ± 4.65 min, and this peak is baseline separated so it does not interfere with the quantification.

**Figure 5 toxins-13-00531-f005:**
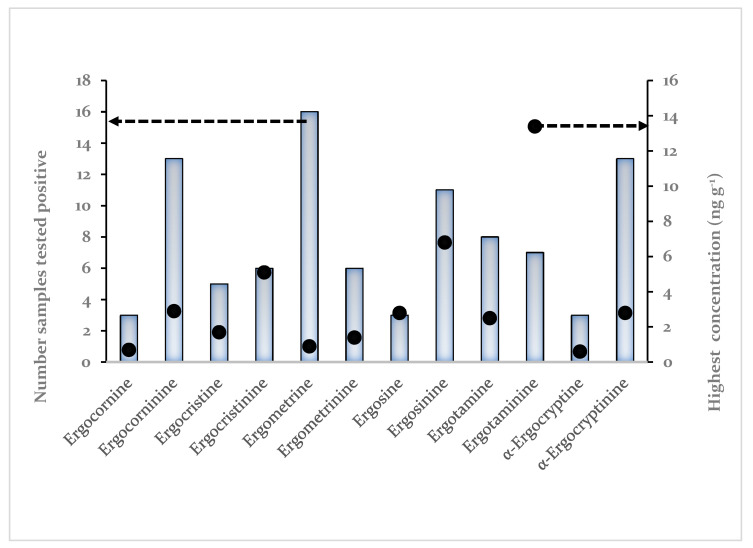
Summary of detection frequency of each ergot alkaloid (left ordinate, bars) and the highest single concentration detected (right ordinate, black dots).

**Figure 6 toxins-13-00531-f006:**
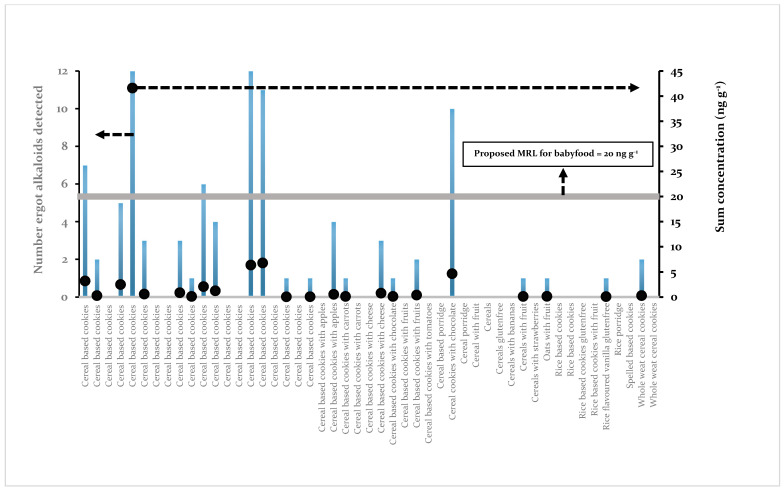
Total number of ergot alkaloids detected (left ordinate, bars) per sample and the total sum of all ergot alkaloids detected in a sample (right ordinate, black dots). The proposed MRL of 20 ng g^−1^ is represented by a horizontal grey bar.

**Table 1 toxins-13-00531-t001:** Overview of the LC-MS parameters i.e., retention time (t_R_), precursor ion, cone voltage, product ions, and collision energy used in the method.

Analytes	t_R_ [min]	Precursor Ion [*m*/*z*]	Cone Voltage [V]	Product Ion *m*/*z*		
				(Collision Energy in eV)		
				Quantifier	1st Qualifier	2nd Qualifier
Ergocornine	4.1	562.4	30	268.1 (25)	223.1 (35)	208.0 (45)
Ergocorninine	4.8	562.4	30	277.1 (25)	223.1 (35)	305.2 (25)
Ergocristine	4.4	610.4	30	223.1 (35)	208.0 (45)	305.1 (25)
Ergocristinine	5.2	610.4	30	223.1 (35)	305.1 (25)	208.0 (45)
Ergometrine	2.0	326.2	35	208.1 (30)	223.1 (25)	181.0 (35)
Ergometrinine	2.7	326.2	35	208.1 (30)	223.1 (25)	181.0 (35)
Ergosine	3.7	548.3	40	223.1 (30)	208.1 (40)	191.9 (50)
Ergosinine	4.5	548.3	40	223.1 (30)	191.9 (50)	208.1 (40)
Ergotamine	3.8	582.3	35	223.1 (35)	208.1 (50)	297.0 (30)
Ergotaminine	4.7	582.3	35	223.1 (35)	297.0 (30)	208.1 (50)
α-Ergocryptine	4.3	576.4	30	223.1 (35)	268.1 (25)	208.0 (45)
α-Ergocyptinine	5.1	576.4	30	223.2 (35)	305.2 (30)	291.2 (25)

**Table 2 toxins-13-00531-t002:** Overview of the validation results i.e., chromatographic retention time (t_R_) in minutes, mean recovery (%), total measurement uncertainty (MU in %), the limit of detection (LOD) calculated by MassLynx as 3.3* S/N ratio, the limit of quantification (LOQ) using 10* S/N ratio, and the reporting limit (RL) used for routine analyses and the regression coefficient R^2^.

	(t_R_)	Rec (%)	MU (%)	LOD (ng g^−1^)	LOQ (ng g^−1^)	RL (ng g^−1^)	R^2^
Ergocornine	4.1	103	32	0.07	0.2	0.5	0.99
Ergocorninine	4.8	110	31	0.07	0.2	0.5	0.99
Ergocristine	4.4	106	48	0.10	0.3	0.5	0.99
Ergocristinine	5.2	111	40	0.07	0.2	0.5	0.99
Ergometrine	2.0	80	50	0.07	0.2	0.5	0.99
Ergometrinine	2.7	87	52	0.03	0.1	0.5	0.99
Ergosine	3.7	103	26	0.07	0.2	0.5	0.99
Ergosinine	4.5	104	30	0.03	0.1	0.5	0.99
Ergotamine	3.8	106	29	0.17	0.5	0.5	0.99
Ergotaminine	4.7	110	48	0.17	0.5	0.5	0.99
α-Ergocryptine	4.3	104	27	0.03	0.1	0.5	0.99
α-Ergocryptinine	5.1	108	28	0.07	0.2	0.5	0.99

**Table 3 toxins-13-00531-t003:** Detailed validation data. Nominal spiked concentration (nom.cc.), average detected concentration (av. meas), bias (accuracy), repeatability (S_rep_), maximum repeatability (2/3 Horwitz), within-laboratory reproducibility (Srw), and maximum within-laboratory reproducibility (Horwitz).

	Nom.cc. (ng g^−1^)	Av.Meas (ng g^−1^)	Bias (%)	Repeatability		Within-Lab Reproducibility	
				S_rep_ (%)	2/3 Horwitz (%	Srw (%)	Horwitz (%)
Ergocornine	0.5	0.5	0%	16	33	18	50
	1	1.03	3%	11	30	11	45
	2.5	2.69	8%	5	26	9	39
Ergocorninine	0.5	0.55	10%	12	33	13	50
	1	1.08	8%	7	30	7	45
	2.5	2.77	11%	7	26	7	39
Ergocristine	0.5	0.54	8%	28	33	31	50
	1	1.04	4%	13	30	13	45
	2.5	2.66	6%	14	26	14	39
Ergocristinine	0.5	0.56	12%	14	33	14	50
	1	1.08	8%	11	30	11	45
	2.5	2.95	18%	11	26	12	39
Ergometrine	0.5	0.42	−16%	23	33	23	50
	1	0.81	−19%	12	30	12	45
	2.5	2.06	−18%	5	26	6	39
Ergometrinine	0.5	0.48	−4%	34	33	34	50
	1	0.89	−11%	17	30	17	45
	2.5	2.29	−8%	7	26	7	39
Ergosine	0.5	0.53	6%	10	33	10	50
	1	1.02	2%	4	30	4	45
	2.5	2.7	8%	7	26	9	39
Ergosinine	0.5	0.5	0%	13	33	15	50
	1	1.02	2%	8	30	9	45
	2.5	2.74	10%	12	26	12	39
Ergotamine	0.5	0.54	8%	10	33	10	50
	1	1.02	2%	10	30	10	45
	2.5	2.72	9%	10	26	11	39
Ergotaminine	0.5	0.57	14%	23	33	25	50
	1	1.05	5%	15	30	15	45
	2.5	2.81	12%	16	26	16	39
α-Ergocryptine	0.5	0.53	6%	7	33	9	50
	1	1.04	4%	7	30	7	45
	2.5	2.77	11%	9	26	9	39
α-Ergocryptinine	0.5	0.54	8%	6	33	6	50
	1	1.09	9%	2	30	8	45
	2.5	2.75	10%	9	26	9	39

**Table 4 toxins-13-00531-t004:** Concentrations (ng g^−1^) of ergot alkaloids in 49 commercial baby food samples and the sum per sample. Blank cells represent values below LOD (see Table 2).

	Ergocornine	Ergocorninine	Ergocristine	Ergocristinine	Ergometrine	Ergometrinine	Ergosine	Ergosinine	Ergotamine	Ergotaminine	α-Ergocryptine	α-Ergocryptinine	SUM
Cereal based cookies		0.3	0.2	0.5				0.7	0.4	0.8		0.3	3.2
Cereal based cookies								0.1		0.2			0.3
Cereal based cookies													
Cereal based cookies		0.2						0.6	0.6	0.8		0.3	2.5
Cereal based cookies	0.7	2.9	1.7	5.1	0.9	1.4	2.8	6.8	2.5	13.4	0.6	2.8	41.6
Cereal based cookies		0.3						0.1				0.2	0.6
Cereal based cookies													
Cereal based cookies													
Cereal based cookies					0.4	0.3						0.1	0.9
Cereal based cookies		0.1											0.1
Cereal based cookies		0.3		0.6	0.2	0.3		0.4				0.3	2.1
Cereal based cookies		0.3			0.2				0.2			0.5	1.3
Cereal based cookies													
Cereal based cookies													
Cereal based cookies	0.2	0.5	0.3	0.9	0.4	0.5	0.5	0.8	0.5	0.8	0.2	0.6	6.4
Cereal based cookies	0.2	0.7	0.2	1.0	0.3	0.5		1.3	0.6	1.2	0.2	0.6	6.8
Cereal based cookies													
Cereal based cookies					0.1								0.1
Cereal based cookies													
Cereal based cookies					0.1								0.1
Cereal based cookies with apples													
Cereal based cookies with apples		0.2			0.1			0.1				0.2	0.5
Cereal based cookies with carrots					0.1								0.1
Cereal based cookies with carrots													
Cereal based cookies with cheese													
Cereal based cookies with cheese		0.3			0.1							0.4	0.8
Cereal based cookies with chocolate					0.1								0.1
Cereal based cookies with fruits													
Cereal based cookies with fruits					0.1				0.3				0.4
Cereal based cookies with tomatoes													
Cereal based porridge													
Cereal cookies with chocolate		0.4	0.2	0.9	0.5	0.5	0.5	0.9	0.2	0.2		0.4	4.7
Cereal porridge													
Cereal with fruit													
Cereals													
Cereals gluten free													
Cereals with bananas													
Cereals with fruit												0.1	0.1
Cereals with strawberries													
Oats with fruit					0.2								0.2
Rice based cookies													
Rice based cookies													
Rice based cookies gluten free													
Rice based cookies with fruit													
Rice flavored vanilla gluten free					0.1								0.1
Rice porridge													
Spelled based cookies													
Whole wheat cereal cookies		0.2						0.1					0.3
Whole wheat cereal cookies													

## Data Availability

Not Applicable.

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
