# Peer review of "A Targeted UHPLC-MS/MS Method Validated for the Quantification of Ergot Alkaloids in Cereal-Based Baby Food from the Belgian Market"

_toxins, 2021, doi:10.3390/toxins13080531_

Round 1

Reviewer 1 Report

The authors have described a new developed, optimized and validated UHPLC-MS/MS method for the simultaneous determination of six ergot alkaloids in 49 cereal-based baby food from the Belgian market.

            In my opinion the manuscript is quite innovative, it is well written, presented and provides details of a method that may become useful in laboratories  for monitoring and quality control of food.

This manuscript can be accepted after minor revision:

line 9: ,,-1” should be superscript

line 99: v/v should be in parentheses

Table1:

column 2: ,,R” should be subscript (tR)

line 126: should be 2.1.2.

line 156: ,,2.3. Method validation” should be from a new paragraph

line 186: there should be a dot at the end of the sentence

Table 2:

tR (min) should be in the second column

units (ng g-1) should be written above (first row) e.g. LOD (ng g-1)

Table 3:

column 2 and 3: ,,-1” should be superscript

column 5: ,,rep” should be subscript

Data shown in Table 3 (column 1 and 2) should be written with dots instead of commas

line 238: should be a pause (…...and the extraction….)

Data shown in Table 4 should be written with dots instead of commas

Sincerely yours,

Wioletta Parys

Author Response

User Menu               

Home (/user/myprofile)

Manage Accounts (/user/manage_accounts)

Change Password (/user/chgpwd)

Edit Profile (/user/edit) Logout (/user/logout)

Submissions Menu 

Submit Manuscript (/user/manuscripts/upload)

Display Submitted Manuscripts (/user/manuscripts/status)

Display Co-Authored Manuscripts (/user/manuscripts/co- authored)

English Editing

Journal      Toxins  (https://www.mdpi.com/journal/toxins) (ISSN 2072-6651) Manuscript ID      toxins-1298379

Type Article Number of Pages 20

Title      A targeted UHPLC-MS/MS method validated for the quantification of ergot alkaloids in cereal- based baby food from the Belgian market

Abstract       Following pending new legislation in the European Union setting a maximum of 20 ng g-1 for the total sum of ergot alkaloids in dry cereal-based baby food, a new UHPLC-MS/MS method was developed. It is suitable for the quantification of six ergot alkaloids, ergocornine, ergocristine, er- gometrine, ergosine, ergotamine, α-ergocryptine, and their corresponding epimers. The method is able to reliably detect individual ergot alkaloids at a level as low as 0.5 ng g-1. The method uses a modified QuEChERS extraction approach before UHPLC-MS/MS analysis. The method showed good sensitivity, accuracy, and precision. It has been applied to 49 samples from the Belgian mar-ket. In 26 samples, not a single ergot alkaloid was detected, in 23 out of 49 samples at least one ergot alkaloid was detected with 2 samples containing 12 ergot alkaloids.

Ergometrine was the alkaloid most frequently detected i.e. 16 out of 49 samples. Only one sample, testing positive for all 12 ergot alkaloids, would be non-conform to the newly proposed MRL.

Author's Reply to the Review Report (Reviewer 1)

(/user/pre_english_article/status)

Discount Vouchers (/user/discount_voucher)

Invoices (/user/invoices)

LaTex Word Count (/user/get/latex_word_count)

* Author's Notes to

Reviewer

Please provide a point-by-point response to the reviewer’s comments and either enter it in the box below or upload it as a Word/PDF file. Please write down "Please see the attachment." in the box if you only upload an attachment. An example can be found here (/bundles/mdpisusy/attachments/Author/Example for author to respond reviewer - MDPI.docx? e9a1de4761e1496f).

File

Edit

View

Insert

Format

Tools

Table

Help

Paragraph

Reviewers Menu     

Reviews (/user/reviewer/status)

Volunteer Preferences (/volunteer_reviewer_info/view)

Word / PDF                                   No file chosen

Submit    or     Save as draft (submit later)

Review Report Form

Open Review      (x) I would not like to sign my review report

( ) I would like to sign my review report

English language

and style

( ) Extensive editing of English language and style required

( ) Moderate English changes required

( ) English language and style are fine/minor spell check required

(x) I don't feel qualified to judge about the English language and style

Yes

Does the introduction provide sufficient

Can be improved

Must be improved

Not applicable

background and include all relevant references?

Is the research design appropriate?

(x)

( )      ( )      ( )

Are the methods adequately described?

(x)

( )      ( )      ( )

Are the results clearly presented?

(x)

( )      ( )      ( )

Are the conclusions supported by the results?

(x)

( )      ( )      ( )

(x)     ( )       ( )       ( )

Comments and Suggestions for

Authors

The authors have described a new developed, optimized and validated UHPLC-MS/MS method for the simultaneous determination of six ergot alkaloids in 49 cereal-based baby food from the Belgian market.

In my opinion the manuscript is quite innovative, it is well written, presented and provides details of a method that may become useful in laboratories for monitoring and quality control of food.

This manuscript can be accepted after minor revision: line 9: ,,-1” should be superscript

Adapted

line 99: v/v should be in parentheses

Adapted

Table1:

column 2: ,,R” should be subscript (tR)

In tR the R is already in subscript but as t is in uppercase and R is in lower case they appear identical in word.

In the beginning of the previous phrase it also shows that at least to the naked eye there both normal case.

I have set “t” in letter size 10 and “R” in letter size 8.

line 126: should be 2.1.2.

Adapted

line 156: ,,2.3. Method validation” should be from a new paragraph line 186: there should be a dot at the end of the sentence

I have added “and”.

Table 2:

tR (min) should be in the second column

I have added tR and again R in a smaller letter size.

units (ng g-1) should be written above (first row) e.g. LOD (ng g-1)

Adapted

Table 3:

column 2 and 3: ,,-1” should be superscript column 5: ,,rep” should be subscript

Data shown in Table 3 (column 1 and 2) should be written with dots instead of commas

Table 3 has been adapted on all 4 points

line 238: should be a pause (…...and the extraction….)

Adapted as: “The conditions chosen for both the preparation of standards and the extraction , i.e. alkaline conditions combined with acetonitrile, should minimize epimerization.”

Data shown in Table 4 should be written with dots instead of commas

All commas have been replaced by points.

Sincerely yours, Wioletta Parys

Submission Date      29 June 2021

Date of this review      05 Jul 2021 09:47:50

© 1996-2021 MDPI (Basel, Switzerland) unless otherwise stated Disclaimer Terms and Conditions (https://www.mdpi.com/about/terms-and- conditions) Privacy Policy (https://www.mdpi.com/about/privacy)

Reviewer 2 Report

This manuscript summarizes a well designed study that provides detailed methods for a sensitive, validated assay that addresses difficult to detect toxin derived from fungi. I see no issues with the work presented, the study design, the methods or the results. The article is concise, well-written, and thoughtful. The assay is robust and more sensitive than the guidelines outlined by the governing agencies. Minor edits are needed for formatting and word choice, otherwise, I see no reason not to accept the manuscript. 

Author Response

Journals  (https://www.mdpi.com/about/journals/)

Information (https://www.mdpi.com/guidelines)        Initiatives

About  (https://www.mdpi.com/about/)

User Menu               

Home (/user/myprofile)

Manage Accounts (/user/manage_accounts)

Change Password (/user/chgpwd)

Edit Profile (/user/edit) Logout (/user/logout)

Submissions Menu 

Submit Manuscript (/user/manuscripts/upload)

Display Submitted Manuscripts (/user/manuscripts/status)

Display Co-Authored Manuscripts (/user/manuscripts/co- authored)

English Editing

Journal      Toxins  (https://www.mdpi.com/journal/toxins) (ISSN 2072-6651) Manuscript ID      toxins-1298379

Type Article Number of Pages 20

Title      A targeted UHPLC-MS/MS method validated for the quantification of ergot alkaloids in cereal- based baby food from the Belgian market

Abstract       Following pending new legislation in the European Union setting a maximum of 20 ng g-1 for the total sum of ergot alkaloids in dry cereal-based baby food, a new UHPLC-MS/MS method was developed. It is suitable for the quantification of six ergot alkaloids, ergocornine, ergocristine, er- gometrine, ergosine, ergotamine, α-ergocryptine, and their corresponding epimers. The method is able to reliably detect individual ergot alkaloids at a level as low as 0.5 ng g-1. The method uses a modified QuEChERS extraction approach before UHPLC-MS/MS analysis. The method showed good sensitivity, accuracy, and precision. It has been applied to 49 samples from the Belgian mar-ket. In 26 samples, not a single ergot alkaloid was detected, in 23 out of 49 samples at least one ergot alkaloid was detected with 2 samples containing 12 ergot alkaloids.

Ergometrine was the alkaloid most frequently detected i.e. 16 out of 49 samples. Only one sample, testing positive for all 12 ergot alkaloids, would be non-conform to the newly proposed MRL.

Author's Reply to the Review Report (Reviewer 2)

(/user/pre_english_article/status)

Discount Vouchers (/user/discount_voucher)

Invoices (/user/invoices)

LaTex Word Count (/user/get/latex_word_count)

* Author's Notes to

Reviewer

Please provide a point-by-point response to the reviewer’s comments and either enter it in the box below or upload it as a Word/PDF file. Please write down "Please see the attachment." in the box if you only upload an attachment. An example can be found here (/bundles/mdpisusy/attachments/Author/Example for author to respond reviewer - MDPI.docx? e9a1de4761e1496f).

File

Edit

View

Insert

Format

Tools

Table

Help

Paragraph

Reviewers Menu     

Reviews (/user/reviewer/status)

Volunteer Preferences (/volunteer_reviewer_info/view)

Word / PDF                                   No file chosen

Submit    or     Save as draft (submit later)

Review Report Form

Open Review      (x) I would not like to sign my review report

( ) I would like to sign my review report

English language

and style

( ) Extensive editing of English language and style required

( ) Moderate English changes required

(x) English language and style are fine/minor spell check required

( ) I don't feel qualified to judge about the English language and style

Yes

Does the introduction provide sufficient

Can be improved

Must be improved

Not applicable

background and include all relevant references?

Is the research design appropriate?

(x)

( )      ( )      ( )

Are the methods adequately described?

(x)

( )      ( )      ( )

Are the results clearly presented?

(x)

( )      ( )      ( )

Are the conclusions supported by the results?

(x)

( )      ( )      ( )

(x)     ( )       ( )       ( )

Comments and Suggestions for

Authors

This manuscript summarizes a well designed study that provides detailed methods for a sensitive, validated assay that addresses difficult to detect toxin derived from fungi. I see no issues with the work presented, the study design, the methods or the results. The article is concise, well-written, and thoughtful. The assay is robust and more sensitive than the guidelines outlined by the governing agencies. Minor edits are needed for formatting and word choice, otherwise, I see no reason not to accept the manuscript.

Submission Date      29 June 2021

Date of this review      23 Jul 2021 21:00:17

© 1996-2021 MDPI (Basel, Switzerland) unless otherwise stated            Disclaimer           Terms and Conditions (https://www.mdpi.com/about/terms-and- conditions)

Reviewer 3 Report

The authors of the manuscript titled “A targeted UHPLC-MS/MS method validated for the quantifi-2 cation of ergot alkaloids in cereal-based baby food from the 3 Belgian market” have presented a very interesting work about ergot alkaloids in cereal-based baby food. The manuscript is well written and presented.

However, there are some minor revision to be clarified (reported below), which should be revised before the publication of the manuscript.

Line 7: Please, use colon after six ergot alkaloids before the list.  

Line 15: Please, specify the full name before the acronym “MRL”.

Line 40: References should be added always in the same way in the manuscript (at the end of the sentence, before the comma).

Line 47: “They have a double bond between either C-8 and C-9 or C-9 and C-10”. Is this correct? Please, verify.

Line 94: “A QuEChERS based method was first applied to ergot alkaloids by Malachova UHPLC-MS/MS”. Do you mean by Malachova and co-workers? Please, remove UHPLC-MS/MS.

Line 99 and 115: Please, write “v/v” in italic form, and correct in the whole manuscript.

Line 151: The authors have reported the modified QuEChERS approach used to purify the sample; however, no information about the protocol were detailed described. Please, could the authors briefly describe the protocol?

Line 156: Please, move “2.3. Method validation” in the line below.

Table 2: Please, write in the first column “Tr (min)” instead only min, as retention times were reported below.

Line 293: maybe the conclusion should be reported at the end of the manuscript, after materials and methods section (unless the journal guidelines suggested to put here).

Author Response

Journals  (https://www.mdpi.com/about/journals/)

Information (https://www.mdpi.com/guidelines)        Initiatives

About  (https://www.mdpi.com/about/)

User Menu               

Home (/user/myprofile)

Manage Accounts (/user/manage_accounts)

Change Password (/user/chgpwd)

Edit Profile (/user/edit) Logout (/user/logout)

Submissions Menu 

Submit Manuscript (/user/manuscripts/upload)

Display Submitted Manuscripts (/user/manuscripts/status)

Display Co-Authored Manuscripts (/user/manuscripts/co- authored)

English Editing

Journal      Toxins  (https://www.mdpi.com/journal/toxins) (ISSN 2072-6651) Manuscript ID      toxins-1298379

Type Article Number of Pages 20

Title      A targeted UHPLC-MS/MS method validated for the quantification of ergot alkaloids in cereal- based baby food from the Belgian market

Abstract       Following pending new legislation in the European Union setting a maximum of 20 ng g-1 for the total sum of ergot alkaloids in dry cereal-based baby food, a new UHPLC-MS/MS method was developed. It is suitable for the quantification of six ergot alkaloids, ergocornine, ergocristine, er- gometrine, ergosine, ergotamine, α-ergocryptine, and their corresponding epimers. The method is able to reliably detect individual ergot alkaloids at a level as low as 0.5 ng g-1. The method uses a modified QuEChERS extraction approach before UHPLC-MS/MS analysis. The method showed good sensitivity, accuracy, and precision. It has been applied to 49 samples from the Belgian mar-ket. In 26 samples, not a single ergot alkaloid was detected, in 23 out of 49 samples at least one ergot alkaloid was detected with 2 samples containing 12 ergot alkaloids.

Ergometrine was the alkaloid most frequently detected i.e. 16 out of 49 samples. Only one sample, testing positive for all 12 ergot alkaloids, would be non-conform to the newly proposed MRL.

Author's Reply to the Review Report (Reviewer 3)

(/user/pre_english_article/status)

Discount Vouchers (/user/discount_voucher)

Invoices (/user/invoices)

LaTex Word Count (/user/get/latex_word_count)

* Author's Notes to

Reviewer

Please provide a point-by-point response to the reviewer’s comments and either enter it in the box below or upload it as a Word/PDF file. Please write down "Please see the attachment." in the box if you only upload an attachment. An example can be found here (/bundles/mdpisusy/attachments/Author/Example for author to respond reviewer - MDPI.docx? e9a1de4761e1496f).

File

Edit

View

Insert

Format

Tools

Table

Help

Paragraph

Reviewers Menu     

Reviews (/user/reviewer/status)

Volunteer Preferences (/volunteer_reviewer_info/view)

Word / PDF                                   No file chosen

Submit    or     Save as draft (submit later)

Review Report Form

Open Review      (x) I would not like to sign my review report

( ) I would like to sign my review report

English language

and style

( ) Extensive editing of English language and style required

( ) Moderate English changes required

( ) English language and style are fine/minor spell check required

(x) I don't feel qualified to judge about the English language and style

Yes

Does the introduction provide sufficient

Can be improved

Must be improved

Not applicable

background and include all relevant references?

Is the research design appropriate?

(x)

( )

( )      ( )

Are the methods adequately described?

( )

(x)

( )      ( )

Are the results clearly presented?

(x)

( )

( )      ( )

Are the conclusions supported by the results?

( )

(x)

( )      ( )

(x)     ( )       ( )       ( )

Comments and Suggestions for

Authors

The authors of the manuscript titled “A targeted UHPLC-MS/MS method validated for the quantifi-2 cation of ergot alkaloids in cereal-based baby food from the 3 Belgian market” have presented a very interesting work about ergot alkaloids in cereal-based baby food. The manuscript is well written and presented.

However, there are some minor revision to be clarified (reported below), which should be revised before the publication of the manuscript.

Line 7: Please, use colon after six ergot alkaloids before the list.

This has been adapted

Line 15: Please, specify the full name before the acronym “MRL”.

This has been adapted

Line 40: References should be added always in the same way in the manuscript (at the end of the sentence, before the comma).

This has been adapted

Line 47: “They have a double bond between either C-8 and C-9 or C-9 and C-10”. Is this correct? Please, verify.

I’m pretty sure this is at least so for the ergot alkaloids discussed here.

Line 94: “A QuEChERS based method was first applied to ergot alkaloids by Malachova UHPLC- MS/MS”. Do you mean by Malachova and co-workers? Please, remove UHPLC-MS/MS.

UHPLC-MS/MS has been removed

Line 99 and 115: Please, write “v/v” in italic form, and correct in the whole manuscript.

It has been adapted at 5 places: lines 99, 118, 159, 349 and 414.

Line 151: The authorshave reported the modified QuEChERS approach used to purify the sample; however, no information about the protocol were detailed described. Please, could the authors briefly describe the protocol?

The following sentence was added : In this modified QuEChERS approach 4g of sample is extrcated for at least 1 hour with a 30 mLsolution of 84% acetonitril and 16% (both in volume, not mass) water containing 200mg L-1.

Line 156: Please, move “2.3. Method validation” in the line below.

This has been adapted

Table 2: Please, write in the first column “Tr (min)” instead only min, as retention times were reported below.

In tR the R is already in subscript but as t is in uppercase and R is in lower case they appear identical in word.

In the beginning of the previous phrase it also shows that at least to the naked eye there both normal case.

I have set “t” in letter size 10 and “R” in letter size 8.

Line 293: maybe the conclusion should be reported at the end of the manuscript, after materials and methods section (unless the journal guidelines suggested to put here).

We did indeed do this following the example-template given on the website

Submission Date      29 June 2021

Date of this review      22 Jul 2021 19:55:15

© 1996-2021 MDPI (Basel, Switzerland) unless otherwise stated            Disclaimer           Terms and Conditions (https://www.mdpi.com/about/terms-and- conditions)
